# Alkaloid Lindoldhamine Inhibits Acid-Sensing Ion Channel 1a and Reveals Anti-Inflammatory Properties

**DOI:** 10.3390/toxins11090542

**Published:** 2019-09-18

**Authors:** Dmitry I. Osmakov, Sergey G. Koshelev, Victor A. Palikov, Yulia A. Palikova, Elvira R. Shaykhutdinova, Igor A. Dyachenko, Yaroslav A. Andreev, Sergey A. Kozlov

**Affiliations:** 1Shemyakin-Ovchinnikov Institute of Bioorganic Chemistry, Russian Academy of Sciences, 117997 Moscow, Russia; osmadim@gmail.com (D.I.O.); sknew@yandex.ru (S.G.K.); yaroslav.andreev@yahoo.com (Y.A.A.); 2Institute of Molecular Medicine, Sechenov First Moscow State Medical University, 119991 Moscow, Russia; 3Branch of the Shemyakin-Ovchinnikov Institute of Bioorganic Chemistry, Russian Academy of Sciences, 6 Nauki Avenue, 142290 Pushchino, Russia; viktorpalikov@mail.ru (V.A.P.); yuliyapalikova@bibch.ru (Y.A.P.); dyachenko@bibch.ru (I.A.D.)

**Keywords:** acid-sensing ion channel subtype 1a, bisbenzylisoquinoline alkaloid, lindoldhamine, nociception, inflammation

## Abstract

Acid-sensing ion channels (ASICs), which are present in almost all types of neurons, play an important role in physiological and pathological processes. The ASIC1a subtype is the most sensitive channel to the medium’s acidification, and it plays an important role in the excitation of neurons in the central nervous system. Ligands of the ASIC1a channel are of great interest, both fundamentally and pharmaceutically. Using a two-electrode voltage-clamp electrophysiological approach, we characterized lindoldhamine (a bisbenzylisoquinoline alkaloid extracted from the leaves of *Laurus nobilis* L.) as a novel inhibitor of the ASIC1a channel. Lindoldhamine significantly inhibited the ASIC1a channel’s response to physiologically-relevant stimuli of pH 6.5–6.85 with IC_50_ range 150–9 μM, but produced only partial inhibition of that response to more acidic stimuli. In mice, the intravenous administration of lindoldhamine at a dose of 1 mg/kg significantly reversed complete Freund’s adjuvant-induced thermal hyperalgesia and inflammation; however, this administration did not affect the pain response to an intraperitoneal injection of acetic acid (which correlated well with the function of ASIC1a in the peripheral nervous system). Thus, we describe lindoldhamine as a novel antagonist of the ASIC1a channel that could provide new approaches to drug design and structural studies regarding the determinants of ASIC1a activation.

## 1. Introduction

Acid-sensing ion channels (ASICs) belong to the family of amiloride-sensitive degenerin/epithelial Na+ channels [1]. ASICs are widely expressed in the peripheral sensory and central neurons, where they serve an important function in the transmission of signals associated with local pH changes, both during normal neuronal activity and in several pathological conditions that cause significant extracellular acidosis [1,2,3].

In mammals, ASICs have six isoforms: 1а, 1b, 2a, 2b, 3, and 4. In the neurons of the central nervous system, ASIC1a, ASIC2, and ASIC4 are the dominant isoforms; these all have a wide distribution in the brain and spinal cord. ASIC3 and ASIC1b are widely distributed in the neurons of the peripheral nervous system, but also have been found in the brain [4]. In total, ASIC1a is one of the widely-expressed primary acid sensors in mammalians in the brain and primary sensory neurons of the dorsal root ganglion [5]. Together with the ASIC3 channel, ASIC1a is the most sensitive to protons with a pH for half-maximal activation values of 6.4–6.7. Activation occurs within a few milliseconds and is accompanied by desensitization lasting several seconds [6,7].

АЅІС1а plays an important role in diverse physiological and pathological processes. This channel is directly involved in synaptic plasticity, learning, nerve-stimulation transmission, fear and anxiety, epilepsy, ischemic processes, neuronal death, and pain sensations [8]. When the central nervous system is under ischemic conditions, ASIC1a is one of the main factors in the dysregulation of ion homeostasis, which leads to neuronal death [2,9]. Sustained activation of ASIC1a leads to an excessive influx of cations, which causes ischemic brain damage. During ischemic diseases, glycolysis is increased, resulting in lactic-acid accumulation and subsequent tissue acidosis. Either the blockade or the transgenic knockdown of ASIC1a induces neuroprotective effects in models of ischemia; thus, ASIC1a is an important target of therapies for ischemic brain damage [10].

A limited number of molecules are known to affect ASIC1a. Endogenous neuropeptides dynorphins, RF-amide peptides, and endogenous cationic polyamine spermine potentiate ASIC1a’s response to acidification by inhibiting the desensitization of the channel, which leads to the neuronal damage in ischemic stroke models [11,12]. Endogenous opioid peptide nocistatin activates ASIC1a at physiological pH and decreases the sensitivity of the channel to the protons [13]. The snake toxin MitTx at nanomolar concentrations activates the ASIC1a channel and induces pain behavior [14]. Other snake toxins known as mambalgins inhibit that channel with a half-maximal effective concentration (IC_50_) of 55 нМ. The spider toxin PcTx1 also inhibits ASIC1a (IC_50_ 1 nM) by promoting the desensitization of the channel. The use of mambalgins and PcTx1 in pain models in vivo demonstrates their significant antinociceptive effect [15,16]. Low molecular weight compounds used in medical practice, such as amiloride (diuretic), diarylamidines (anti-infective drugs), ibuprofen (a non-steroidal anti-inflammatory drug), and chloroquine (the anti-malarial drug) also inhibit ASIC1a when applied at high micromolar concentrations [3,17].

Medicinal herbs have been of therapeutic interest for centuries and have resulted in considerable contributions to the development of new drugs. Medicinal plants are useful sources for the isolation of novel, biologically-active molecules and the subsequent synthesis, modification, and optimization of those molecules’ biological properties [18]. Alkaloids are a diverse natural group of biologically-active, secondary metabolites; alkaloids have attracted much attention due to their toxic and medicinal properties. One of these interesting and promising alkaloids is the dendrogenins, which elicit the proliferation of neuronal cells [19]. The other perspective compound, a plant alkaloid, sinomenine, extracted from the roots of *Sinomenium acutum*, effectively inhibits ASIC1a (IC_50_ ~1 μM) and possesses anti-inflammatory, antinociceptive, and neuroprotective effects [20]. However, there are no data on the effect of this compound on other ASIC isoforms. Here, we describe a plant bisbenzylisoquinoline alkaloid named lindoldhamine (LIN) as a potent inhibitor of ASIC1a activation by physiologically-relevant acidification. LIN was extracted from the leaves of *Laurus nobilis* belonging to the family Lauraceae. Representatives of this family synthesize an abundant variety of alkaloids with 22 structurally-distinct moieties (more comprehensive information can be found in the review [21]). Scholars have described LIN as an acetylcholinesterase inhibitor with an IC_50_ value of 3.5 μM [22]. LIN has demonstrated trypanocidal activity due to inhibition of trypanothione reductase (IC_50_ 27 μM) [23]. At a concentration of 100 μM, LIN completely inhibits platelet aggregation induced by collagen, arachidonic acid, and platelet-activating factor [24]. It has recently been shown that LIN acts as a proton-independent activator of ASIC3 (with ЕC_50_ values of 1.5 mM and 3.2 mM for human and rat ASIC3 channels, respectively). Besides, LIN potentiated the transient component of the human, but not rat ASIC3 current and restored it from desensitization, with EC_50_ values of 3.7 and 16.2 μM, respectively [25].

To assess the pharmacological potential of LIN, we tested it on the ASIC1a channel. We found that LIN stimulus-dependently inhibited the acid-induced currents of ASIC1a and produced anti-inflammatory effects in mice. These results demonstrated a good correlation between in vitro and in vivo data, as well as the participation of ASIC1a in some pathological processes occurring in the organism.

## 2. Results

### 2.1. LIN Proton-Dependently Inhibits the ASIC1a Current

Bisbenzylisoquinoline alkaloid LIN was extracted from the leaves of *Laurus nobilis* as described previously [25] (Figure 1A,B). Using two-electrode voltage-clamp recordings from *Xenopus laevis* oocytes that express rat ASIC1a, we found that, at the holding membrane potential of −50 mV, an LIN concentration of 300 μM effectively inhibited ASIC1a during channel activations with mild acidic stimuli (pH above 6.0). LIN inhibited ASIC1a by 98.1 ± 0.3%, 93.7 ± 0.9%, and 69.3 ± 2.3% for pHs of 6.85, 6.7, and 6.5, respectively (Figure 1C). LIN’s inhibitory power tended to decrease as the pH dropped from 7.4 to 6.0 (34 ± 4% inhibition), and the inhibition was non-significant for stimuli with a pH less than 5.5.

To evaluate the effectiveness of LIN’s ASIC1a inhibition, we measured the dose-dependence of LIN’s inhibitory effect with various activating stimuli (Figure 2).

As expected, LIN very effectively inhibited ASIC1a during mild acidic stimulations, and its effectiveness (IC_50_ value) significantly decreased as the acidity of the stimulus increased. An important observation is that the stable dose-response curve shifted from a sub-micromolar IC_50_ value to one measuring hundreds of micromoles for pH values below 6.0. When applied before the pH dropped to above 6.0, LIN was able to inhibit the current completely; however, an application before a pH 6.0 stimulus dramatically changed LIN’s mode of action, causing the inhibitory effect to reach saturation at 52 ± 11%. For a stimulus with the more acidic pH of 5.5, the dose-dependent curve no longer fit well. The values of the dose-dependent parameters are summarized in Table 1.

### 2.2. LIN’s Effect on the pH Dependence of the ASIC1a Activation

In the presence of 0.3 mM of LIN, the proton dependence of the ASIC1a activation demonstrated an acidic shift (Figure 3). The amplitude’s fit with the F_2_ equation (see the formula in the Section 4 Materials and Methods) indicated that LIN shifted both the pH_50_1 and pH_50_2 values towards a more acidic area, with pH_50_1 shifting from 6.72 ± 0.02 for the control to 6.47 ± 0.03 with LIN (*p* < 0.001) and with pH_50_2 shifting from 6.49 ± 0.04 for the control to 6.23 ± 0.09 with LIN (*p* < 0.05).

The Hill coefficient n_H_ showed a strong tendency to decrease in the presence of LIN, although this change had no statistical significance (6.6 ± 0.5 for the control and 5.7 ± 0.7 for 0.3 mM LIN (*p* = 0.14)).

### 2.3. LIN Produced an Anti-Inflammatory Effect, but Failed to Decrease the Response to Acid

We studied the anti-inflammatory effect using the complete Freund’s adjuvant (CFA)-induced inflammation model. The intraplantar injection of CFA in the hind paw caused hyperalgesia (i.e., increased sensitivity to noxious mechanical and thermal stimuli) and swelling due to the inflammatory process. We chose a 1 mg/kg dose so that the LIN would reach a blood plasma concentration of about 40 µM, as estimated with regard to blood volume (~8% of weight) and blood plasma content (55–65% of that volume [26]).

The group of mice treated with saline 24 h after the CFA injection showed significantly higher paw diameter (3.26 ± 0.03 mm) relative to that of the control group (2.09 ± 0.03 mm), which we treated with saline twice. Intravenous administration of LIN at a dose of 1 mg/kg decreased paw edema significantly, relative to that of the control group by 25%, 21%, and 19% 2, 4, and 24 h after administration, respectively (Figure 4A).

CFA injection provoked a significant decrease of inflamed hind paw withdrawal latency to thermal stimuli (11.44 ± 0.59 s) in comparison with the group treated with saline twice (19.34 ± 1.56 s). Intravenous administration of LIN (1 mg/kg) reversed 83.4% of thermal hyperalgesia. The latency of the response to the thermal stimulus for the LIN-treated group was 18.03 ± 1.18 s (Figure 4B).

The intraperitoneal injection of acetic acid provoked a pain behavior in the mice, as characterized by abdominal contractions (acetic acid-induced writhes). LIN pretreatment had no significant analgesic effect on the mice (Figure 4C), as the number of writhes was 34.4 ± 2.9 for the saline control group and 36.2 ± 2.4 for the LIN pretreated group.

## 3. Discussion

Bisbenzylisoquinoline alkaloids represent a fairly large family of isoquinoline alkaloids, in the abundance occurring in plants of the Lauraceae family. The pharmacology of bisbenzylisoquinoline alkaloids is also quite extensive and has remarkable medicinal relevance. Most of them are potent anticancer agents and possess anti-inflammatory, antiplasmodial, and antiviral activity. The most important and widely-studied compounds are primarily tetrandrine, curine, and curine-related compounds, cepharanthine, cycleanine, fangchinoline, and neferine [27]. For example, tetrandrine selectively inhibited the alpha7 subtype of neuronal nicotinic acetylcholine receptors and possessed an anti-inflammatory, antifibrotic, anticancer, and neuroprotective properties [28,29], Bisbenzylisoquinoline alkaloid lindoldhamine (LIN) has not been widely studied yet. To date, LIN has been shown to inhibit acetylcholinesterase, trypanothione reductase, platelet aggregation, as well as to activate and potentiate the ASIC3 channel [22,23,24,25].

In this study, we show the action LIN on the ASIC1a channel in vitro, as well as its effect in two pain models in vivo. We found that LIN effectively inhibited ASIC1a currents activated by mild acidic stimuli. LIN’s inhibition efficiency correlated with the strength of the applied acid stimulus. For the minimal activation stimulus of pH 6.85, LIN completely eliminated the current, with an IC_50_ value of 9 μM. LIN also completely inhibited the current for two other low-scale stimuli (pHs 6.7 and 6.5), but its inhibitory effectiveness was more than 10-times lower. Medium acidification (pH 6.0 or lower) produced a greater ion current that the LIN could not completely inhibit; therefore, as the activation stimuli increased, the maximum inhibition decreased, and the IC_50_ value increased. We measured the inhibitory effect under pre-incubation conditions when the ligand had a preference for the channel binding, so we can assume that the LINs’ binding site at least partially overlaps with some of the channel’s numerous protonation sites.

A fast pH drop is essential to open the ASIC1a channel, and the rate of acidification was very fast in our experiments. Therefore, we were able to realize LIN’s inhibitory effect through the competition for overlapping protonation sites. LIN was most efficacious for weak, physiologically-relevant acidification (Figure 2 and Figure 3), but when the acidic stimuli had a pH below 6.0, the protons could simultaneously affect approximately 16 binding sites [30,31], resulting in a conversion of the ion channel into its so-called pre-activation form [32]. The LIN likely could not bind with the pre-activated form of ASIC1a, thus causing it to lose its affinity and its inhibitory activity such that it only affected a small population of the channels still not be pre-activated. As a result, LIN’s overall effect was to shift the proton dependence curve of ASIC1a activation towards a more acidic region (Figure 3). Other well-known ASIC1a inhibitors, including 2-guanidine-4-methylquinazoline (GMQ) and mambalgins, have the same effect [33,34].

For rat ASIC1a, some researchers have described proton activation as having non-sigmoidal dependence [35,36] that fit well by equation F_2_ (Materials and Methods). This equation describes the simultaneous protonation of a pool of “highly-cooperative” sites (pH_50_1) and additional “non-cooperative” site (pH_50_2) within the channel. LIN significantly decreased both the pH_50_1 and pH_50_2 parameters by 0.25 units, which indicated that it equally impacted both the highly-cooperative sites and the non-cooperative site. We thus assumed that LIN influenced more than one protonation site in ASIC1a, either through the overlap of binding sites or LIN could affect these sites allosterically. However, to date, there are not enough data to verify with which protonation sites within ASIC1a LIN interacts.

We evaluated LIN’s effects in vivo to confirm that it is an ASIC1a inhibitor. We used two tests: the acetic acid-induced writhing test (i.e., strong external acidification) and the CFA-induced inflammation test (i.e., inflammation-associated acidification).

In the CFA-induced inflammation test, LIN had a significant anti-inflammatory effect, as it reduced both thermal hyperalgesia and paw edema. Although the ASIC1a channel is widely represented in the neurons of the central nervous system and plays a significant role in central nociception in the neurons of the dorsal root ganglion, ASIC1a expression is elevated during inflammation and upregulated by inflammatory mediators [37,38]. In addition, in the neurons of the spinal dorsal horn, the ASIC1a channel contributed to inflammatory hypersensitivity to pain [39]. During the inflammation process, the pH of the inflamed area was 0.5–1 units lower than the physiological pH [40,41]; this amount is significant and is sufficient to activate the ASIC1a that are most sensitive to protons [6]. LIN’s effectiveness confirms that ASIC1a plays an important role in CFA-induced inflammation; the data for this model correlated well with the in vitro data, which showed that LIN completely inhibited the channel at stimuli with a pH from 6.85–6.5.

In the acetic acid-induced writhing test, LIN did not have an analgesic effect. This may be because LIN cannot inhibit the ASIC1a channel for stimuli with pH below 6.0. An alternative assumption is that the ASIC1a channel does not participate in the development of acid-induced pain processes, confirming what was reported earlier [42].

LIN and other isoquinoline compounds [25,43] potentiated human ASIC3, but did not affect rat ASIC3; therefore, the animal acetic acid-induced writhing model could not produce the combined effect of ASIC3 potentiation and ASIC1a inhibition. In humans, the combined effect on ASIC1a and ASIC3 could have one of two results. First, the opposite effects could counteract this, resulting in zero final effect; this has been shown for high doses of APETx2, a well-known ASIC3 antagonist [44,45]. The other option is the significant enhancement of the analgesic effect because of desensitizing action on neurons of potentiators or weak activators, as was previously shown on the modulators of transient receptor potential vanilloid 1 (TRPV1) and transient receptor potential ankyrin 1 (TRPA1) channels [46,47,48,49].

ASIC1a’s hyperactivity is associated with pathological processes in the central nervous system. Either a knockout or a pharmacological blockade of ASIC1a prevents the development of certain neurodegenerative diseases [8]. When combined with the fact that LIN effectively inhibits acetylcholinesterase activity and thus is a therapeutic option for preventing the progression of neurodegenerative diseases [22], this leads to a new perspective in which LIN can be used as a multipotent drug in the treatment of neurodegenerative diseases.

In summary, we demonstrated that LIN effectively inhibited ASIC1a activity under mild acidosis, but that this effectiveness decreased significantly as the acidity of the stimulus increased. These results are in good agreement with those for in vivo pain models. LIN effectively demonstrated anti-inflammatory activity, but did not have an analgesic effect against acetic acid-induced pain. These results confirm the importance of the ASIC1a channel in the development of pain processes during inflammation and together with LIN’s previously-reported activity open new perspectives for the development of drugs to treat neurodegenerative diseases.

## 4. Materials and Methods

### 4.1. Chemical Reagents and Compounds

Lindoldhamine isolated from dried leaves of *Laurus nobilis* was used in all experiments. We obtained LIN using the isolation procedure [25] described previously. All reagents were obtained from Sigma-Aldrich (Steinheim, Germany). A working solution of the ligand was prepared in the ND96 buffer (96 mM NaCl, 2 mM KCl, 1 mM MgCl_2_, and 5 mM HEPES (4-(2-hydroxyethyl)-1-piperazineethanesulfonic acid) titrated to pH 7.4 with NaOH) immediately before the experiments.

### 4.2. Ethics Statement

This study strictly complied with the World Health Organization’s International Guiding Principles for Biomedical Research Involving Animals. All experiments were approved by the Institutional Policy on the Use of Laboratory Animals of the Shemyakin-Ovchinnikov Institute of Bioorganic Chemistry Russian Academy of Sciences (Protocol Number 267/2018; date of approval: 2 October 2018) and by the Institutional Commission for the Control and Use of Laboratory Animals of the Branch of the Shemyakin-Ovchinnikov Institute of Bioorganic Chemistry of the Russian Academy of Sciences (identification code: 688/19; date of approval: 10 January 2019).

### 4.3. Electrophysiological Experiments 

Electrophysiological experiments were performed on *X. laevis* oocytes expressing rat acid-sensing ion channel isoform 1a (ASIC1a channel). Unfertilized oocytes were harvested from female *X. laevis*. Tricaine methanesulfonate (MS222) (0.17% solution) was used to anaesthetize frogs, which, after surgery, were kept in a separate tank until they had completely recovered from the anesthesia. The follicle cell layers were removed from the oocytes by treatment at room temperature for 1.5–2 h with 1 mg/mL of collagenase in ND96 medium (96 mM NaCl, 2 mM KCl, 1 mM MgCl_2_, and 5 mM HEPES (4-(2-hydroxyethyl)-1-piperazineethanesulfonic acid) titrated to pH 7.4 with NaOH) lacking calcium. Selected healthy Stage IV and V oocytes were injected with 2.5–5 ng of cRNA, synthesized from PCi plasmids containing the rat ASIC1a isoform, using the Nanoliter 2000 microinjection system (World Precision Instruments, USA). The injected oocytes were kept for 48–72 h at 17 °C and then for up to 6 days at 15 °C in ND96 medium supplemented with an antibiotic (gentamycin, 50 μg/mL). Two-electrode voltage-clamp recordings were carried out at a holding potential of −50 mV using the GeneClamp500 amplifier (Axon Instruments, Inverurie, U.K.). Microelectrodes were filled with 3 M KCl. The data were filtered and digitized at 20 Hz and 100 Hz, respectively, using the L780 AD converter (L-Card, Moscow, Russia). The external bath solution was ND96; the pH was adjusted to 7.4. For the activating test ND96 solutions with 6.5 ≤ pH ≤ 7.0 and pH <6.5, HEPES was replaced by 10 mM MOPS (3-(N-morpholino)propanesulfonic acid) and 10 mM MES (2-(N-morpholino)ethanesulfonic acid), respectively. A computer-controlled valve system was used to achieve a streamflow of about 1 mL/min and a solution exchange rate about 60 mL/min in the recording chamber.

### 4.4. In Vivo Assay

#### 4.4.1. Animals

Male adult CD-1 mice (20–25 g, 31 mice) were maintained at a normal 12-h light-dark cycle with water and food available ad libitum.

#### 4.4.2. Complete Freund’s Adjuvant-Induced Inflammation and Thermal Hyperalgesia

The oil/saline (1:1) CFA emulsion was injected into the dorsal surface of the left hind paw of mice (20 μL/paw) 24 h before the measurement, which caused the development of inflammation and thermal hyperalgesia of the paw. Saline (20 μL) was injected into control mice. The inflamed paw withdrawal latencies to thermal stimulation (53 °C) were measured 30 min after the LIN or saline injection. The paw diameter was evaluated using a digital caliper before the CFA injection, before LIN and saline administration, and 2, 4, and 24 h after the administration.

#### 4.4.3. Acetic Acid-Induced Writhing (Abdominal Constriction Test of Visceral Pain)

Mice were divided into separate groups, and acetic acid in saline (0.6%, 10 mL/kg) was injected intraperitoneally 30 min after the intravenous administration of the LIN or saline for the control group. Mice were directly located inside transparent glass cylinders, and the number of writhes was registered for 30 min.

### 4.5. Data and Statistical Analysis

Analysis of electrophysiological data was performed using OriginPro 8.6 software (OriginPro 8.6.0 (32 bit) version, OriginLab Corporation, Northampton, MA, USA, 2011). Curves were fitted using the following logistic equations:(a) F_1_(x) = ((a1 – a2)/(1 + (x/x0)^n_H_)) + a2(1)
where F_1_(x) is the response value for a given LIN concentration; x is the concentration of LIN; a1 is the control response value (fixed at 100%); a2 is the response value at maximal inhibition (% of the control); x0 is the IC_50_ value; and n_H_ is the Hill coefficient;
(b) F_2_(x) = A/((1 + (x/[pH_50_1])^n_H_) × (1 + (x/[pH_50_2]))(2)
where [pH_50_1] is the half-maximal concentration of protons binding to the highly-cooperative sites (or half probability of “high-cooperative” sites’ occupancy); [pH_50_2] is the half-maximal concentration of protons binding to the non-cooperative site (or half probability of “non-cooperative” site occupancy); n_H_ is the Hill coefficient; and A is the current’s maximum amplitude.

The data significance in animal tests was determined by the analysis of variance (ANOVA) followed by Tukey’s post-hoc test. Data are presented as the mean ± SEM.

## Figures and Tables

**Figure 1 toxins-11-00542-f001:**
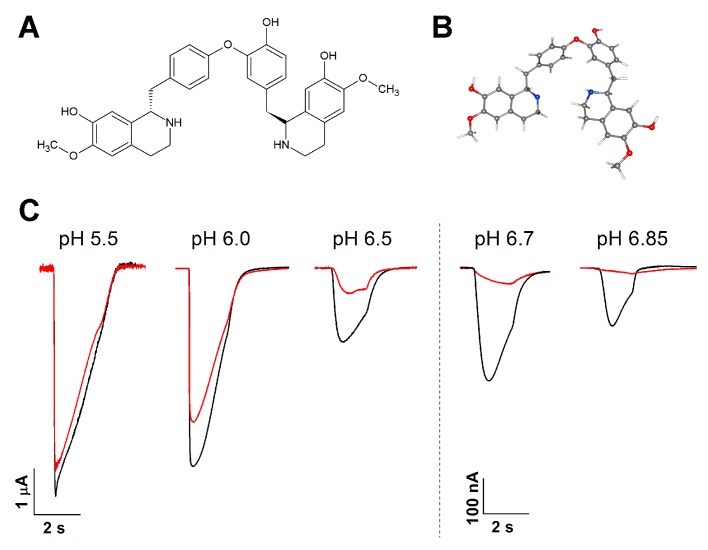
Lindoldhamine (LIN)’s effects on rat ASIC1a. (**A**) LIN’s chemical structure. (**B**) LIN’s predicted (cambered and flex) 3D structure (from PubChem; CID 10370752). Carbon atoms are in grey, hydrogen atoms in white, oxygen atoms\in red, and nitrogen atoms in blue. (**С**) The inhibitory effect that LIN (300 μM) had on the ASIC1a channel’s activation by fast external solution acidification from pH 7.4 to the corresponding pH stimulus. The black traces are the control current; the red traces are the current obtained after 15 s of LIN pre-application. All presented traces are for a single cell.

**Figure 2 toxins-11-00542-f002:**
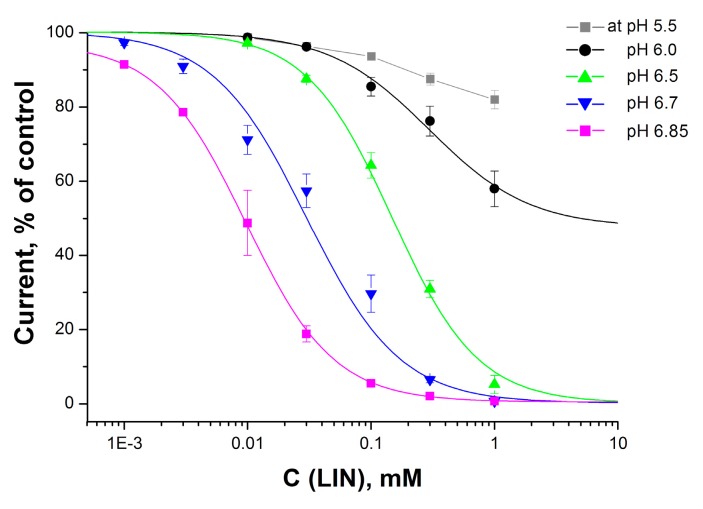
Dose-response curves for LIN’s inhibitory activity on ASIC1a currents. The colored curves denote the corresponding pH stimuli that activate the channel, relative to the conditioning pH of 7.4. Each point is the mean ± SEM of five measurements. The data were fitted using the F_1_ logistic equation (see the Materials and Methods Section for details).

**Figure 3 toxins-11-00542-f003:**
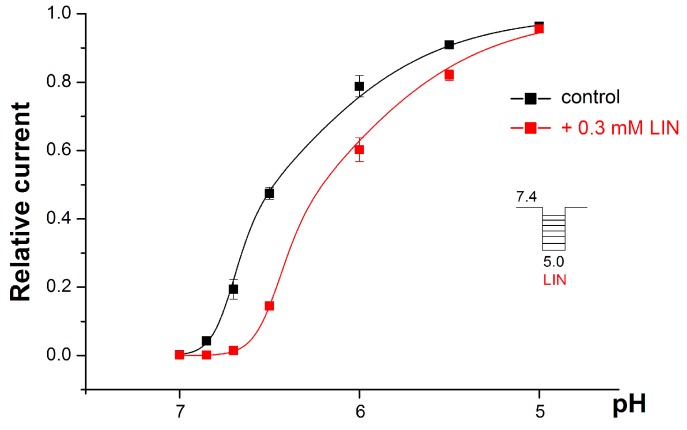
The pH dependence of the ASIC1a activation by protons alone (black line) and with the co-application of protons and 0.3 mM of LIN (red line). The ASIC1a channel was held at pH 7.4 and then activated by various acidic stimuli. The relative current is the amplitude of the peak current that the acidic stimuli evoked, normalized to the maximum amplitude (which we calculated for each cell via individual fitting). The data are presented as the mean ± SEM (*n* = 5).

**Figure 4 toxins-11-00542-f004:**
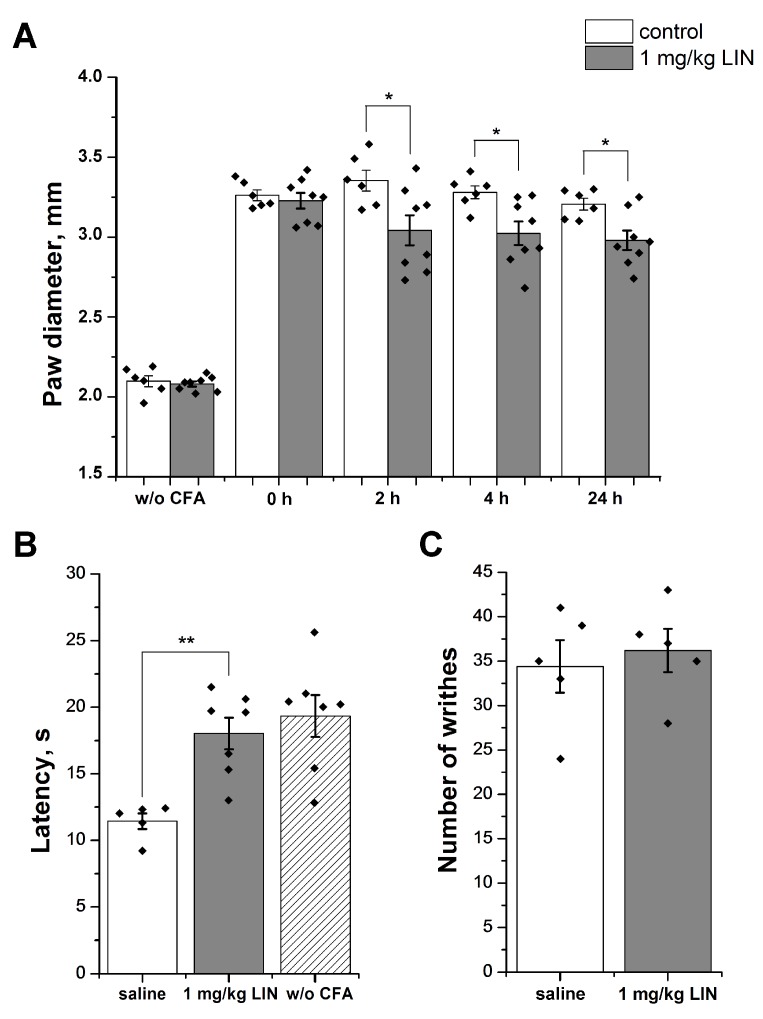
LIN activity in animal models. (**A**) LIN’s anti-inflammatory effect. We induced paw edema via complete Freund’s adjuvant (CFA) injection and estimated values before the administration of the CFA and the testing compound. LIN (1 mg/kg, i.v.) significantly reduced edema, even 24 h after injection. (**B**) LIN’s reversal of thermal hyperalgesia. LIN (1 mg/kg, i.v., 30 min before testing) significantly reversed CFA-induced thermal hyperalgesia and prolonged the withdrawal latency for an inflamed hind paw placed on a hot plate. (**C**) LIN’s effect on the writhing test. Pretreatment of mice with LIN (1 mg/kg, i.v., 30 min before testing) did not have any significant effect on the results of the writhing test, which involved the intraperitoneal administration of acetic acid. The results are presented as the mean ± SEM (*n* = 5–8); the *p*-values of the LIN group vs. the saline group are based on an analysis of variance and on Tukey’s test. * *p* < 0.05, ** *p* < 0.01.

**Table 1 toxins-11-00542-t001:** Calculated IC_50_ and Hill coefficient (nH) values of LIN’s inhibitory action for activating stimuli, by pH.

	pH 6.85	pH 6.7	pH 6.5	pH 6.0
**IC_50_, µM**	9 ± 0.3	25 ± 6	150 ± 10	296 ± 146
**n_H_**	1.22 ± 0.03	1.29 ± 0.12	1.25 ± 0.05	1.07 ± 0.15

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
