# Peer review of "Alkaloid Lindoldhamine Inhibits Acid-Sensing Ion Channel 1a and Reveals Anti-Inflammatory Properties"

_toxins, 2019, doi:10.3390/toxins11090542_

Round 1

Reviewer 1 Report

This is a short research report to describe LIN, an alkaloid from leaves of laurus nobilis which demonstrated to inhibit acid-induced currents of ASIC1a channel in mild acidosis and could then be a potential antagonist of ASIC1a channel. The following suggestions should be followed for the revision.

This manuscript did not mention how the LIN concentration of 300 uM was determined, which could effectively inhibit ASIC1a channel. Has this concentration been optimized yet? Why did the authors choose a dose of 1 mg/kg of LIN for in vivo study? The authors estimated that the dose of 1 mg/kg would reach to the serum concentration of 40 uM based on blood volume. It might not be the case since LIN might accumulate more in some organs and less in the blood circulation. The authors should have measured the LIN concentration in serum. The dose of LIN for the in vivo study should have been optimized. M & M section seems not fully completed yet. For examples, lines 227 and 252 “…”. What is the sequence similarity of ASIC between rats and mice? Why did the authors choose rat ASIC for in vitro assay and mouse ASIC for in vivo study? The discussion between lines 194 and 206 is confusing. It should be reworded. “dose-dependence curve”, “dose-dependence parameters” should read “dose-dependent curve”, “dose-dependent parameters” At line 92, “pH drop above 6” should read “pH dropped to above 6”. What does it mean by “by used logistic equation” at line 96? The manuscript suffers from grammatical and editorial errors and some of them are listed below. “The” in the title should be omitted. “bv” should read “by” in the title. ASIC should be spelled out the first time it appears in the abstract. At line 7, “a most” should be “the most”. At lines 9 and 10, “both fundamentally and medicinal” should read “both fundamentally and pharmaceutically”. Errors in the use of singular and plural nouns. For example, at line 20, “ASIC1a channels” should read “ASIC1a channel”. Verb tense errors. For example, at line 190, “ data, which show” should read “data, which showed”. Verb tense agreement problems in complex sentences. The tense of the main clause determines the tenses in the subordinate clause. However, in this manuscript, there are quite a few of complex sentences (lines 63 to 64, 148 to 149, 213 to 214, etc.), where the verb of the main clause is paste tense while that of the subordinate clause is present tense.

Listed are only some grammatical errors in the manuscript. The manuscript should be re-read carefully and checked for all errors and reworded for some phrases.

Reviewer 2 Report

The paper should be revised and written in an organized manner so that the readers could understand. The following points should be considered.

1.      Make the abstract more informative.

2.      Please make the Introduction more comprehensive. In line 33, you can make figure on isoforms so we could understand. The introduction needs more background about the whole study or paper.

3.      Page 2, lines 46 to 51. Expand this. I think you need connectivity between sentences and paragraphs.

4.      Did you do EC50 or dose response with pH 6.85? Also, what is the reason it is best at pH 6.85?

5.      Enzymes in our body are active on approximate pH 6.8 to 7.4 something. How did you correlate on this and how did you choose pH in your research? Did you get just randomly?

6.      I think the title is not suited to the content well.

7.      I think the authors should revise the paper in an organized manner. The last part was not written well with unfinished sentences.

Reviewer 3 Report

12-Augest-2019
Journal:  Toxins.

Title: The alkaloid lindoldhamine inhibits ASIC1a 2 activation by mild acidification and possesses anti-inflammatory properties.

Dear Editor,

The authors have presented lindoldhamine (LIN) as a novel antagonist of acid-sensing ion channels (ASIC1a) providing a new approach to drug design and structural studies. LIN also, exhibited significant anti-inflammatory effect in Complete Freund’s Adjuvant-Induced Inflammation assay. The manuscript is adequate for publication after some minor revisions as indicated below.

Comments to Authors:

Title:

Please avoid the abbreviations in the title. Authors could rewrite the title to be more concise. Please add the authors' names and affiliations.

Abstract:

Authors could add high quality graphical abstract in accordance to journal instructions. Authors would design the abstract in a structural form according to journal instructions. Authors would mention the scientific name of the used plant.

Key words:

Authors wouldn`t use abbreviation in the keywords list such as (ASIC1). Authors would add and select more specific key words. In key contribution: Bisbenzylisoquinoline is the (LIN), so please remove the coma.

Introduction:

Please mention the plant source of lindoldhamine (LIN). Please could you add the known knowledge about laurel leaves. Could you indicate the other biological importance of this compound? Please add indications of А-317567 and GMQ in line 50. Authors could check the writing style through out of the manuscript. Authors could benefit from this reference in the introduction:

Khalifa, S.A., De Medina, P., Erlandsson, A., El-Seedi, H.R., Silvente-Poirot, S. and Poirot, M., 2014. The novel steroidal alkaloids dendrogenin A and B promote proliferation of adult neural stem cells. Biochemical and biophysical research communications, 446(3), pp.681-686.

Materials and Methods

Authors would clarify the number of mice used in in vivo In the two lines 234 and 247: please add a description of HEPES. Authors would add identification of the abbreviation (ASIC). Authors would clarify if they use the extract or lindoldhamine (LIN) in the experiment. Please explain the isolation methods of lindoldhamine (LIN).

Results and discussion:

In figure 2, please mention the used dose of LIN. Please add the indications for TRPV1 and TRPA1 in line 206. Could you determine the difference between (pH501) and (pH502)?

References:

The authors would check the style of writing references in accordance to journal instructions.

Reviewer 4 Report

Acid-sensing ion channels (ASIC) are present in almost all neurons and sense reduced levels of extracellular pH. They are potential drug targets for treating a wide variety of conditions linked to both the CNS and PNS. This manuscript focuses on characterization of lindoldhamine (LIN) as a novel inhibitor of the ASIC1a channel. The authors report that mode of action of LIN differs from other known antagonists of ASIC1a, and postulate that it could provide new approaches to drug design and structural studies regarding the determinants of ASIC1a activation. In my opinion, this is globally a nice and original piece-of-work. It appears to be scientifically sound, some few points need attention before publication. The manuscript is well-written and clear. However, the authors should present more extensively the current knowledge about plant alkaloids especially lindoldhamine and its activity, and better develop on the hypothesis presented in the Introduction; the objectives of this study should be more clearly stated.

Minor comments:

Line 60: change IC50 to IC50

Line 61: as above

Line 170: change These equation describes to This equation describes

Line 270: Tukey’s post-hoc

In the M&M section authors should state from which plant (which part of the plant) the LIN was isolated.

Mentioning the origin of lindoldhamine in the title would be appropriate, as it is not synthetic compound but one isolated from plant.

Round 2

Reviewer 2 Report

Accept